

# Vibrations on mastoid process alter the gait characteristics during walking on different inclines

Yuxiao Sun[1], Dongqi Zhu[1], Huiyan Song[2] and Jung H. Chien[3]

[1] College of Allied Health Professions, University of Nebraska Medical Center, Omaha, Nebraska, United States
[2] Department of Rehabilitation Medicine, West China Hospital, Sichuan University, Chengdu, Sichuan, China
[3] Independent Researcher, Omaha, Nebraska, United States

Corresponding author
Jung H. Chien,
drjc.science@gmail.com

## ABSTRACT

**Background:** Eighty-eight percent of the persons with bilateral vestibular dysfunction have reported at least one fall within the past 5 years. The apparent alternations due to the bilateral vestibular dysfunctions (BVD) are the gait characteristics, such as slower walking speed, prolonged stance phase, and shorter step length. Unexpectedly, due to the prevalence of this BVD being relatively low, attention is not obtained as same as in other vestibular disorders. Moreover, how does walking on different inclines, part of daily activities, alter the gait characteristics under the unreliable bilateral vestibular systems? Previous studies used vibration-based stimulations (VS) as a perturbation to understand the postural control during walking while the bilateral vestibular systems were perturbed. Therefore, this study attempted to extend the knowledge to understand the alternations in spatial-temporal gait characteristics under perturbed bilateral vestibular systems while walking on different inclines.

**Methods:** Nineteen healthy young adults participated in this study. Eight walking conditions were randomly assigned to each participant: 0%, 3%, 6%, and 9% grade of inclines with/without VS. The preferred walking speed was used for gait analysis. The dependent variables were stance time, double support time, step length, step time, step width, foot clearance, and respective variabilities. All dependent variables were defined by two critical gait events: heel-strike and toe-off. *Pre-Hoc* paired comparisons with Bonferroni corrections were used to prioritize the dependent variables. A two-way repeated measure was used to investigate the effect of VS and the effect of inclines on the selected dependent variables from *Pre-Hoc* analysis. *Post-Hoc* comparisons were also corrected by the Bonferroni method.

**Results:** The step length, step time, foot clearance, and foot clearance variability were selected by the *Pre-Hoc* analysis because the corrected paired t-test demonstrated a significant VS effect ($p < 0.05$) on these gait parameters at least one of four inclines. The significant interaction between the effect of VS and the effect of inclines was found in step length ($p = 0.005$), step time ($p = 0.028$), and foot clearance variability ($p = 0.003$). The results revealed that implementing a VS increased step length and step time when walking on 0%, 3%, and 9% of grade inclines. In particular, the foot clearance variability was found when walking on 9% of grade inclines.

**Conclusion:** The observations in the current study suggested that VS increased the step length, step time, foot clearance, and foot clearance variability while walking on inclines. These results suggested that these gait parameters might be promising targets for future clinical investigations in patients with BVD while walking on different inclines. Importantly, the increases in spatial-temporal gait performance under bilateral VS might be an indicator of gait improvement while walking on different inclines.

## INTRODUCTION

The vestibular system is crucial to maintaining the dynamic balance in human locomotion (*Horak, 2009*). Specifically, semicircular canals, utricles, and saccule in the labyrinth play essential roles in detecting the rotational and linear acceleration of body movement (*Rabbitt, 2019*), such as the daily gait. A study reported that persons with bilateral vestibular dysfunctions (BVD) have an increased 31-fold potential fall risk compared to the average residents in the United States; moreover, these patients with BVD have nine times greater fall risk compared to those health controls with dizziness and unsteadiness during standing and walking (*Ward et al., 2013*). One of many factors leading to falls in patients with BVD may be attributed to oscillopsia (*Batuecas-Caletrio et al., 2020*). The oscillopsia is an illusion of the surroundings moving caused by an impairment of the vestibular-ocular reflex, which plays a critical role in correcting eye position during alternations in head position so that vision remains on the target (*Hain, Cherchi & Yacovino, 2018*). In particular, a study involving 37 patients with BVD reported that 81% of these patients had moderate to severe oscillopsia severity while walking at different speeds (*Guinand et al., 2012*). It is worth mentioning in the abovementioned study that 10 out of 37 patients with BVD were not able to walk at 6 km/h (an average and comfortable walking speed for healthy controls) due to the imbalance caused by the oscillopsia (*Guinand et al., 2012*). Therefore, the alternations in gait characteristics in patients with BVD during walking may possibly be caused by this illusion.

A 2008 National Health Interview Survey estimated that a prevalence of 28 per 100,000 or 64,046 US adults had suffered the BVD. Unexpectedly, because the prevalence of this BVD is relatively low, attention is not obtained (*Kim & Kim, 2022*) as same as other vestibular disorders, such as benign paroxysmal positional vertigo, vestibular neuronitis, and Meniere's disease. Several common clinical diagnoses have been developed to identify the patients with BVD, such as the head impulse test (*Halmagyi & Curthoys, 1988*), dynamic visual acuity (*Demer, Honrubia & Baloh, 1994*) and caloric testing, which Dr. Robert Barany invented. The impulse test is to test the eye movements after examiners quickly and unpredictably move the head to 10 to 15 degrees of neck rotation while subjects sit or stand naturally. In patients with BVD, the vestibular-ocular reflex may be impaired or absent, and the eye movements deviate from the designated target toward the

side where the head is rotated when the head impulse test is applied (*MacDougall et al., 2009*). Similar to the head impulse test, dynamic visual acuity is used to determine the accuracy of eye-track ability (vestibular-ocular reflex). At the same time, examiners oscillate the patient's head horizontally or vertically and instruct the patient to read the optotypes on a visual acuity chart. When applying this dynamic visual acuity to patients with BVD, these patients show a decline of two or more lines because the vestibular-ocular reflex may no longer be able to stabilize the gaze on a visual acuity chart (*Sargent et al., 1997*). The caloric test is to examine the integrity of horizontal semicircular canals and the afferent pathway pouring either cold or warm water into ears. A study revealed that patients with BVD reduced caloric responses and underwent rotational chair testing (*Sargent et al., 1997*). Although these abovementioned diagnoses have been widely used in clinics, a review stated that using these diagnoses for identifying mild BVD remains a diagnostic challenge (*Petersen, Straumann & Weber, 2013*). Additionally, a book chapter emphasized that the caloric test has a high sensitivity to unilateral vestibular disorder but is relatively insensitive to BVD (*Youmans & Winn, 2011*).

Based upon the objective of public health in the World Health Organization (WHO) to assess and monitor the health of communities and populations at risk, it is essential to develop an objective measurement to identify BVD. Therefore, in the past decades, increasing studies have attempted to understand the balance control mechanism in different kinds of vestibular dysfunctions (*Chen et al., 2021*; *Herssens et al., 2021*; *Kingma et al., 2019*; *McCrum et al., 2019*) using different measures such as the galvanic vestibular stimulation. A review (*Dlugaiczyk, Gensberger & Straka, 2019*) indicated that galvanic stimulation could be used for diagnosing different types of vestibular deficits by perturbing the vestibular system. Also, this galvanic stimulation can be used for rehabilitation in balance control while turning the amplitude of stimulations to a noisy (sub-threshold, persons can't perceive) level. However, this galvanic vestibular stimulation has its inevitable side effects, such as increases in anxiety levels (*Pasquier et al., 2019*) and mild uncomfortable sensations (*Utz et al., 2011*). These side effects may hinder the true outcomes of motor adjustments due to the unreliable vestibular system. Therefore, there is a need to find an effective measure to identify the alternations in gait characteristics with perturbed bilateral vestibular function. Applying bilateral supra-threshold vibration-based stimulation (persons can perceive) on the mastoid processes is one of the options to perturb the vestibular system and further investigate the role of the vestibular system for balance control in standing and walking (*Chien, Mukherjee & Stergiou, 2016*; *Lin et al., 2022*; *Lu, Xie & Chien, 2022*). In other words, by measuring the increased sway variability and margins of stability induced by vibration-induced stimulations, the different types of vestibular deficits may potentially be identified by the alternations in gait performance (*Chien, Mukherjee & Stergiou, 2016*; *Lu, Xie & Chien, 2022*). Additionally, this vibration-induced stimulation has a similar effect as the galvanic vestibular stimulation without uncomfortable sensations (*Lin et al., 2022*).

The gait characteristics have been used to identify the vestibular function (*Chae, Song & Kim, 2021*; *Herssens et al., 2021*). For example, a longitudinal observation indicates that immediately after having unilateral vestibular neuritis, the step width in these patients is

significantly greater than in controls (*Chae, Song & Kim, 2021*). After 8 weeks of recovery, significant increases in walking speed, stride length, and swing phase and significant decreases in stance phase and step width are observed. These results suggest the feasibility of use in gait characteristics to identify the differences between controls and patients with unilateral vestibular neuritis and also to monitor the progress of recovery. Similarly, when comparing the gait characteristics among healthy controls, patients with bilateral vestibular loss demonstrated shorter stride lengths while walking straight for 10 m. Similarly, a study found higher cadence and lower step time in patients with bilateral vestibulopathy than in controls (*Herssens et al., 2021*).

The gait variability measure has been widely used to identify the potential fall risk in the older population (sample size: 52) (*Hausdorff, Rios & Edelberg, 2001*), and the different types of neurological disorders, such as patients with Parkinson's disease (sample size: 51) (*Ma et al., 2022*) and patients with stroke (sample size: 16) (*Zukowski et al., 2019*). Indeed, the gait variability measure identifies the differences in spatial-temporal gait variabilities between healthy young controls and patients with bilateral vestibulopathy (*McCrum et al., 2019*). However, this gait variability measure failed to determine the differences between healthy older controls and patients with bilateral vestibulopathy (12 older adults, age: 71.5 years and 44 patients with bilateral vestibulopathy, age: 57.6 years) (*McCrum et al., 2019*) in cadence variability, step time variability, step length stability, step width variability, and double support phase variability (*McCrum et al., 2019*). These insignificant results may be attributed to the limited number of measured steps or gait cycles. Or the differences in age and sample sizes between healthy older controls and patients with bilateral vestibulopathy. Or the gait variability itself is not sensitive enough to detect the effect of deteriorated vestibular system on locomotion. Furthermore, *Lu, Xie & Chien (2022)* use the gait variability measure and find the differences in gait variabilities when the vestibular system is perturbed either unilaterally or bilaterally. The results reveal that applying either bilateral or unilateral vestibular vibrations increases the step width variability, which is a sign of relative gait instability, compared to no vestibular stimulation condition, confirming the feasibility of use in variability measure for the vestibular dysfunctions (*Lu, Xie & Chien, 2022*).

Walking on different inclines is part of daily activities. Importantly, walking on different inclines might induce alternations in self-referenced coordinates with respect to global coordinates that could influence vestibular input due to the different perceptions of gravity (*Cromwell, 2003*). Specifically, when walking on the inclines, the vector of a sum of translational acceleration and gravitational acceleration is different from the level walking, resulting in alterations in otolith information (*Dai et al., 1994*). Thus, do these alternations in the inclines affect the gait or gait variability while the bilateral vestibular system becomes unreliable?

To answer all the abovementioned research questions, this study attempted to understand the fundamental alternations in gait characteristics under unreliable bilateral vestibular systems while walking on different inclines in healthy young adults. This study hypothesized that (1) applying the VS on bilateral vestibular systems caused increases in stance time, double support time, step time, and step width but decreased the step length

and foot clearance, and (2) applying the VS increased the gait variability in all gait characteristics.

## MATERIALS AND METHODS

### Participants

Nineteen healthy young adults participated in this study (10 males and nine females; $24.42 \pm 2.11$ years old; the preferred walking speed: $1.41 \pm 0.22$ m/s; body mass: $60.68 \pm 11.42$ kg; height: $1.66 \pm 0.08$ m). We excluded participants if they had any neurological disorders, neuropathy due to diabetes, previous ankle, knee, hip injuries or surgeries, or unanticipated falls in the prior year. Importantly, these participants were also excluded if they got a score above zero on the dizziness handicap inventory, indicating potential deficits in the vestibular system. Also, these participants verbally declared that they had no deficits in the vestibular systems at or before the day of data collection. Moreover, these participants never experienced any type of vestibular stimulation. If participants agreed to attend the data collection, they needed to sign an informed consent before the data collection. This study obeyed the guideline and regulations of the University of Nebraska Medical Center Institutional Review Board that approved this study (IRB# 379-17-EP).

### Experimental materials

The gait characteristics were captured by a Qualisys motion capture system. This system contained eight high-speed infra-red digital cameras (Qualisys AB, Gothenburg, Sweden) and used the Qualisys Tracker Manager (QTM) software (Qualisys AB) to record the three-dimensional gait data at 100 Hz. Four retro-reflective markers were placed on the toe (second metatarsophalangeal joint) and the heel of both legs. Eight inclined treadmill walking conditions (0%, 3%, 6%, and 9% grade of inclination with/without bilateral vestibular perturbation) were randomly assigned to all participants. Also, the RTM 600 treadmill in the current study (Biodex RTM 600; Biomex Medical System, Inc, Shirley, NY, USA; Fig. 1) included a $50.8 \times 160$ cm Teflon-impregnated running deck and incorporated a shock-absorbing surface. The RTM 600 treadmill offered a speed range of 0–10 mph (0–4.47 m/s) with 0.1 mph (0.045 m/s) speed increments. This treadmill also could be configured for 0% to 15% grade (0 to approximately 8.5 degrees) inclines with 1% grade increments. A handrail was provided for participants to maintain their balance if they felt unsteady during walking. Importantly, a safety lanyard and a red safety button were provided for participants and experimenters to stop the treadmill when encountering any potential risks. The safety lanyard was attached to the participants' waists. If participants displayed an unsteady gait and led to the detachment of the safety lanyard, the treadmill was shut off by control panel of treadmill to prevent falls.

The different types of vestibular simulations were generated by a mechanical vibration using two electromechanical vibrotactile transducers (EMS2 tactors; Engineering Acoustics, Casselberry, FL, USA; Fig. 1). These two transducers were attached inside a customized swim cap using double-sided adhesive strips and can be adjusted to place on the mastoid processes bilaterally. These transducers were designed for mounting with a cushion and could produce high displacement levels that allow the vibration to be easily

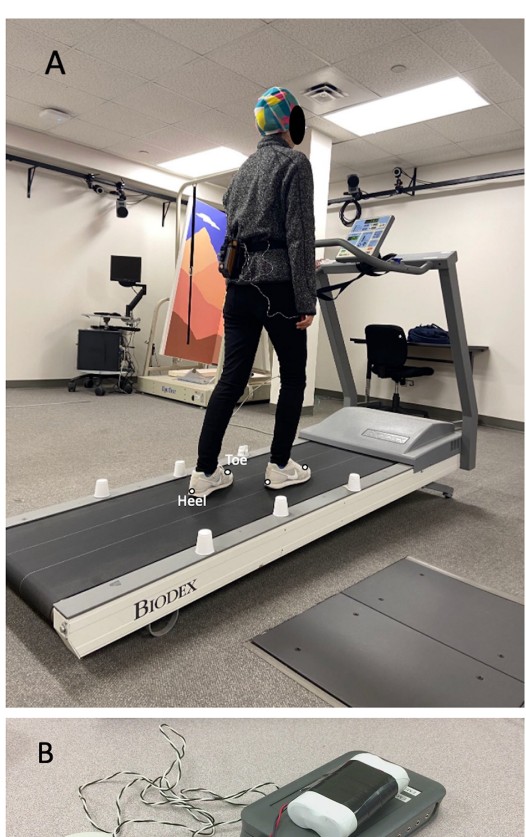

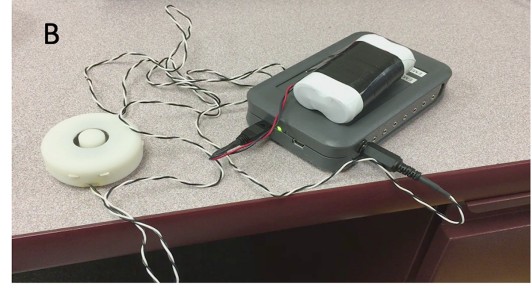

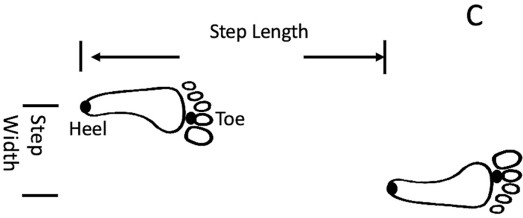

**Figure 1 The diagram of experimental design.** (A) Example session showing walking on a treadmill wearing a vibration system. (B) Vibration system (a controlled unit, two vibrators, and a portable battery); (C) Step length. Four retro-reflective markers were placed on the toe (second metatarsophalangeal joint) and the heel of both legs. One heel-strike was defined as the farthest position that a heel marker could reach in the anterior-posterior direction within one gait cycle, whereas the one toe-off was defined as the minimum position that a toe marker could make in the anterior-posterior direction within one gait cycle. The step time of one gait cycle was defined as the time between the one heel strike and the subsequent heel strike in the contralateral leg. The stance phase was defined as the percentage of the gait cycle from one heel-strike to one toe-off in the same leg. The double support time was defined as the percentage of the gait cycle that both feet stayed on the ground during walking. The step length was defined as the traveling distance of the treadmill belt in the anterior-posterior direction from one heel strike to another contralateral heel strike. Also, the step width was calculated by the distance between two continuous heel strikes in the medial-lateral direction.

sensed even through layers of padding. The maximum peak-to-peak displacement was 2 mm. The height and weight of the tactors were 19.05 mm and 24 g. The diameter of this transducer was 49.26 mm (Fig. 1). The frequency was set at 100 Hz. Also, the amplitude of supra-threshold vibrations was set at 130% of the amplitude the participants could perceive (*Lu, Xie & Chien, 2022*). This frequency of bilateral VS on the mastoid processes was selected as 100 Hz because this frequency is strong enough to trigger the nystagmus and needs compensatory responses from the vestibular system in healthy young adults (*Perez, 2003*), patients with vestibular neuritis (*Nuti & Mandala, 2005*) and in patients with otosclerosis (*Manzari & Modugno, 2008*). The frequency of amplitude of the mastoid vibrations was controlled by software (TAction Creator; Engineering Acoustics, Casselberry, FL, USA) by sending the designed signal from the laptop to the controller through Bluetooth technology. The minimum perceived amplitude was detected by adjusting the amplitude of vibration through TAction Creator commercial software until participants could perceive it during quick standing. This abovementioned procedure was performed three times to get the average minimum perceived amplitude. The vibrations were administrated to participants on both mastoid processes simultaneously. The vibration activation was an impulse type, indicating a 0.5 s activation period and 0.5 s de-activation period. The rationale for using this impulse-type vibration was to reduce the saturation of the vestibular sensation (*Chien, Mukherjee & Stergiou, 2016*).

## Experimental protocol

A total of eight walking trials were randomly assigned to participants in one visit. These walking conditions were walking on 0%, 3%, 6%, and 9% grade of inclines with/without bilateral mastoid vibrations. Each walking trial lasted 3 min. Before the beginning of the data collection, the preferred walking speed was determined for each participant. First, participants needed to stand on the sidebar of the treadmill; then, the belt was accelerated to 0.8 m/s. Next, participants stepped on the treadmill belt, holding the handrail to prevent tripping (Fig. 2). After 30 s, participants were encouraged to walk naturally without holding the handrail. Then, experimenters asked each participant, "is this walking speed similar to the walking speed when you walk on the street?". The walking speed was repeatedly assessed after 10 s (±0.1 m/s) until participants admitted, "this is my comfortable walking speed". After the preferred walking speed was determined, participants needed to walk for 5 min to familiarize the treadmill on 0% grade incline. After the familiarization, participants needed to take a 2-min mandatory rest break for recovery. Then, eight walking conditions were randomly provided to these participants. Also, participants needed to take a 2-min mandatory rest between trials to eliminate the training effect from different inclines and vibrations (*Lu, Xie & Chien, 2022*). Additionally, at the end of each trial, participants were verbally asked whether they had any discomfort sensation, such as nausea, vomiting, or dizziness induced by the inclines or the mastoid vibrations. If they felt any abovementioned discomfort sensations, the data collection was terminated.

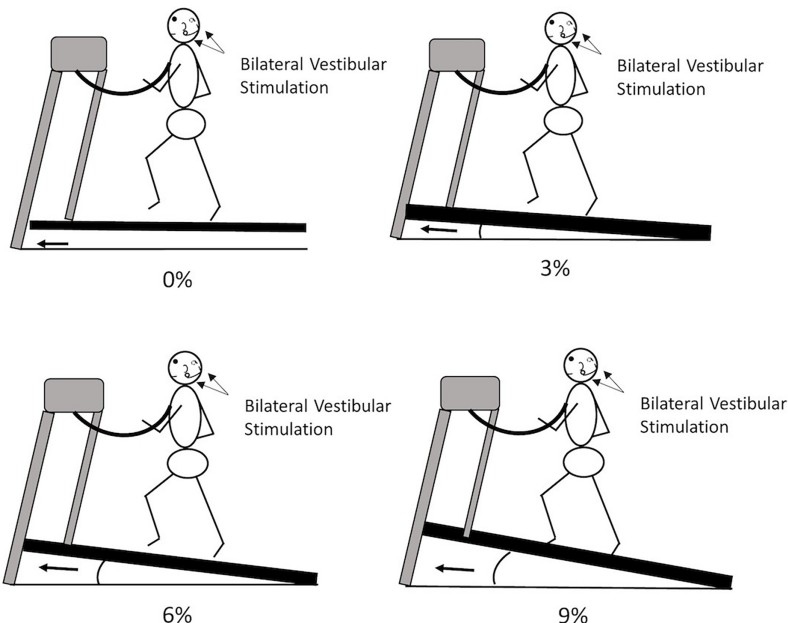

**Figure 2 The experimental protocol.** Participants were randomly assigned to walk on 0%, 3%, 6%, or 9% with/without vibration-based vestibular stimulation.

## Data analysis

The spatial-temporal gait parameters were used as follows: stand time (% of the gait cycle), double support time (% of the gait cycle), step length (mm), step time (ms), step width (mm), foot clearance (mm) and their respective variabilities. The 100 gait cycles were selected from the 31[st] stride to the 130[th] stride to eliminate the large step-to-step fluctuations due to the speed up and slowing down at the beginning/end of the walking. One heel-strike was defined as the farthest position that a heel marker could reach in the anterior-posterior direction within one gait cycle, whereas the one toe-off was defined as the minimum position that a toe marker could make in the anterior-posterior direction within one gait cycle (*Ihlen et al., 2012*). The foot clearance was defined as the maximum vertical height that a heel maker could reach from the ground in one gait cycle (*Mariani et al., 2012*). The step time of one gait cycle was defined as the time between the one heel strike and the subsequent heel strike in the contralateral leg. The stance phase was defined as the percentage of the gait cycle from one heel-strike to one toe-off in the same leg. The double support time was defined as the percentage of the gait cycle in that both feet stayed on the ground during walking. The step length was defined as the traveling distance of the treadmill belt in the anterior-posterior direction from one heel strike to another contralateral heel strike. Also, the step width was calculated by the distance between two continuous heel strikes in the medial-lateral direction. The gait variability was defined as the standard deviation within 100 gait cycles for each dependent variable. The current study selected the standard deviation as a gait variability measure because using the coefficient of variation tended to amplify the values if the mean values were close to zero (*Gouelle et al., 2018*).

## Statistical analysis

A Shapiro–Wilk normality test was used to identify the normality of each spatial-temporal gait parameter and their respective variabilities. The alpha value was set at 0.05. If the alpha value for the Shapiro–Wilk test was greater than 0.05, indicating the data was normally distributed. In the current study, a *pre-hoc* comparison was used for a couple of reasons as follows: (1) using a *pre-hoc* comparison may lower the numbers of multiple comparisons and further reduce the statistical type-I errors, and (2) using a *pre-hoc* comparison can accentuate that only meaningful/interesting hypotheses were included (*Midway et al., 2020*). Thus, in the current study, a paired t-test with Bonferroni correction was used to compare the effect of bilateral VS on each gait parameter while walking on 0%, 3%, 6%, and 9% grades of inclines. If any significant effect of VS was found on each gait characteristic in any incline, this specific gait characteristic was included, and a two-way repeated measure was used (two types of vibrations × four different inclines) for this selected dependent variable. *Post-hoc* multiple comparisons were also corrected by the Bonferroni method. All statistical analysis was performed using SPSS 20 (IBM, Armonk, New York, USA). The way that SPSS calculated the *pre-* and *post-hoc* comparisons using the Bonferroni method was to take the uncorrected *p*-value and multiply it by the number of comparisons made (https://www.ibm.com/support/pages/calculation-bonferroni-adjusted-p-values, last viewed on December 10, 2022). Here, there were a total of twenty-eight multiple comparisons for each selected gait characteristic. Therefore, to obtain the corrected *p*-value, the uncorrected *p*-value needed to be multiplied by twenty-eight. If the corrected *p*-value is less than 0.05, the significance level was reached. If the alpha value for the Shapiro–Wilk test was less than 0.05, indicating the data was not normally distributed, a Friedman test was used. Wilcoxon Signed Rank Test was used for *post-hoc* comparisons for each dependent value. This study used the current sample size based on a previous publication to investigate the effect of mastoid vibrations on the net center of pressure (*Chien, Mukherjee & Stergiou, 2016*). In this abovementioned study, recruiting 20 healthy young could obtain observed power of approximately 1. Additionally, this study used G* power (URL: http://www.gpower.hhu.de/) to calculate the power. The $\eta2 = 0.09$ was used because this specific partial eta squared value was between 0.059 for the moderate effect size and 0.138 for the large effect size based on the partial eta squared method (*Lu, Xie & Chien, 2022*). By this calculation, 80% of statistical power, which represented a reliable balance between the alpha and beta risk, could be reached by recruiting 18 healthy young adults for the repeated measure. In this study, the partial eta squared was used to measure the observed power.

# RESULTS

## Normality tests

The alpha values of the Shapiro–Wilk test for each spatial-temporal gait parameter and individual variability were greater than 0.05, indicating that the data were normally distributed. Therefore, a two-way repeated measure was used for selected spatial-temporal gait parameters and selected variabilities.

### Pre-hoc tests

*Pre-hoc* comparisons revealed that a significance level was reached (VS *vs.* No VS) in step length ($p = 0.004$ for 0% grade walking, $p = 0.013$ for 3% grade walking, $p = 0.002$ for 9% grade walking), step time ($p = 0.027$ for 0% grade walking, $p = 0.028$ for 3% grade walking, $p = 0.028$ for 9% grade walking), step time variability ($p = 0.05$ for 3% grade walking), foot clearance ($p = 0.008$ for 9% grade walking), and foot clearance variability ($p = 0.022$ for 0% grade walking, $p = 0.003$ for 3% grade walking, $p = 0.001$ for 9% grade walking) (Table 1).

### Post-hoc tests

#### Interaction between the effect of inclines and mastoid vibrations on each spatial-temporal gait parameters and variabilities

A significant interaction was found in step length ($F_{3,54} = 4.701$, $p = 0.005$) and step time ($F_{3,54} = 3.283$, $p = 0.028$) (Fig. 3). *Post-hoc* comparisons indicated that significantly greater step length ($p = 0.004$, $p = 0.014$, and $p = 0.034$), and significantly greater step time ($p = 0.027$, $p = 0.028$, and $p = 0.028$) when walking with bilateral mastoid vibrations than when walking without vibrations on 0%, 3% and 9% grade of incline respectively. However, when walking on the 6% grade of incline, the effect of mastoid vibration could not be found in all spatial-temporal gait parameters. The only significant interaction between the effect of inclines and mastoid vibrations on gait variability was the foot clearance variability ($F_{3,54} = 5.207$, $p = 0.003$). Moreover, the *post-hoc* comparisons demonstrated that when applying the bilateral mastoid vibrations, the foot clearance variability was significantly greater during walking on a 3% ($p = 0.003$) and 9% ($p < 0.001$) grade of inclines compared to walking without vibrations.

#### Effect of mastoids vibration on each spatial-temporal gait parameters and variabilities

A significant effect of VS was found in step length ($F_{1,18} = 19.551$, $p < 0.001$), step time ($F_{1,18} = 16.760$, $p = 0.001$), foot clearance ($F_{1,18} = 23.081$, $p < 0.001$), step time variability ($F_{1,18} = 6.110$, $p = 0.024$), and foot clearance variability ($F_{1,18} = 38.485$, $p < 0.001$). Marginal means indicated that implementing a bilateral mastoid vibration significantly increased step length, step time, foot clearance, step time variability, and foot clearance variability.

#### Effect of inclines on each spatial-temporal gait parameters and variabilities

A significant effect of inclines was found in step length ($F_{3,54} = 6.620$, $p = 0.001$), step time ($F_{3,54} = 4.345$, $p = 0.008$), foot clearance ($F_{3,54} = 58.312$, $p < 0.001$), and foot clearance variability ($F_{3,54} = 45.606$, $p < 0.001$). Marginal means demonstrated significantly greater step length, step time, and foot clearance when walking on a 9% grade of incline than when walking on a 0% grade of incline.

#### Effect size

The Partial eta squared values were 0.207 for step length and 0.153 for step time. These values indicated that the effect size was large based on a previous study, meaning that at least 0.138 for a large effect size, 0.059 for a moderate effect size, and 0.01 for a small effect size.

**Table 1 *Pre-hoc* Test.** This table demonstrated the effect of bilateral vestibular stimulation on each gait characteristic while walking on 0%, 3%, 6%, and 9% grades of inclines. This *pre-hoc* tests were used to identify the priorities from multiple dependent variables. A paired t-test with Bonferroni correction was used to compare the effect of bilateral VS on each gait parameter while walking on 0%, 3%, 6%, and 9% grades of inclines. The gait characteristics were as follows: stance time (StanceTime, %), stance time variability (StanceTimeV, %), double support time (DoubleSupport, %), double support time variability (DoubleSupportV, %), step length (StepLength, mm), step length variability (StepLengthV, mm), step time (StepTime, ms), step time variability (StepTimeV, ms), step width (Stepwidth, mm), step width variability (StepwidthV, mm), foot clearance (Footclearance, mm), foot clearance variability (FootclearanceV, mm). No, Walking on the treadmill without bilateral vestibular stimulation; VS, walking on the treadmill with bilateral vestibular stimulation. NS, the *p*-value is larger than 0.05, indicating not significant.

| StanceTime (%) | 0% of grade | 3% of grade | 6% of grade | 9% of grade | StanceTimeV (%) | 0% of grade | 3% of grade | 6% of grade | 9% of grade |
|---|---|---|---|---|---|---|---|---|---|
| No | 63.56 (1.27) | 63.49 (1.13) | 63.62 (1.15) | 63.62 (1.03) | No | 1.62 (0.39) | 1.65 (0.33) | 1.75 (0.44) | 1.78 (0.65) |
| VS | 63.57 (1.20) | 63.61 (1.32) | 63.37 (1.39) | 63.01 (1.95) | VS | 1.75 (0.39) | 1.79 (0.41) | 1.87 (0.57) | 1.86 (0.69) |
| *Pre hoc* (*p* value) | NS | NS | NS | NS | *Pre Hoc* (*p* value) | NS | NS | NS | NS |
| Doublesupport (%) | 0% of grade | 3% of grade | 6% of grade | 9% of grade | DoubleSupportV (%) | 0% of grade | 3% of grade | 6% of grade | 9% of grade |
| No | 25.12 (2.54) | 24.99 (2.27) | 25.23 (2.31) | 25.25 (2.08) | No | 1.44 (0.39) | 1.41 (0.35) | 1.49 (0.37) | 1.49 (0.49) |
| Vs | 25.13 (2.41) | 25.23 (2.65) | 24.75 (2.77) | 24.48 (2.62) | VS | 1.44 (0.38) | 1.47 (0.38) | 1.55 (0.59) | 1.52 (0.57) |
| *Pre hoc* (*p* value) | NS | NS | NS | NS | *Pre Hoc* (*p* value) | NS | NS | NS | NS |
| Steplength (mm) | 0% of grade | 3% of grade | 6% of grade | 9% of grade | StepLengthV (mm) | 0% of grade | 3% of grade | 6% of grade | 9% of grade |
| No | 537.27 (57.27) | 536.82 (54.56) | 546.55 (57.88) | 547.88 (61.44) | No | 134.73 (30.83) | 129.95 (34.73) | 141.94 (40.07) | 141.77 (45.11) |
| Vs | 551.02 (65.72) | 554.57 (65.94) | 552.42 (65.84) | 559.15 (68.91) | VS | 147.78 (26.66) | 146.87 (31.52) | 152.22 (35.69) | 153.18 (46.22) |
| *Pre hoc* (*p* value) | 0.004 | 0.013 | NS | 0.002 | *Pre Hoc* (*p* value) | NS | NS | NS | NS |
| Steptime (ms) | 0% of grade | 3% of grade | 6% of grade | 9% of grade | StepTimeV (ms) | 0% of grade | 3% of grade | 6% of grade | 9% of grade |
| No | 619.25 (55.81) | 617.46 (49.98) | 626.74 (59.73) | 628.25 (58.61) | No | 15.89 (4.40) | 15.29 (4.88) | 17.02 (5.44) | 17.23 (6.31) |
| Vs | 631.39 (57.41) | 635.24 (59.53) | 632.09 (59.36) | 639.64 (58.15) | VS | 17.41 (4.35) | 17.49 (4.75) | 18.07 (4.89) | 18.37 (6.27) |
| *Pre hoc* (*p* value) | 0.027 | 0.028 | NS | 0.028 | *Pre Hoc* (*p* value) | NS | 0.05 | NS | NS |
| Stepwidth (mm) | 0% of grade | 3% of grade | 6% of grade | 9% of grade | StepWidthV (mm) | 0% of grade | 3% of grade | 6% of grade | 9% of grade |
| No | 537.27 (57.27) | 536.82 (54.56) | 546.55 (57.88) | 547.88 (61.44) | No | 14.88 (2.78) | 13.89 (3.71) | 16.15 (2.79) | 15.51 (3.38) |
| Vs | 551.02 (65.72) | 554.57 (65.94) | 552.42 (65.84) | 559.15 (68.91) | VS | 15.05 (2.83) | 15.21 (2.94) | 15.86 (3.27) | 15.60 (3.32) |
| *Pre hoc* (*p* value) | NS | NS | NS | NS | *Pre Hoc* (*p* value) | NS | NS | NS | NS |
| Footclearance (mm) | 0% of grade | 3% of grade | 6% of grade | 9% of grade | FootClearanceV (mm) | 0% of grade | 3% of grade | 6% of grade | 9% of grade |
| No | 231.98 (30.79) | 243.32 (33.02) | 253.18 (32.94) | 260.40 (34.52) | No | 35.96 (20.12) | 39.32 (20.05) | 44.75 (23.52) | 51.98 (28.11) |
| Vs | 235.01 (32.22) | 246.48 (30.59) | 256.58 (32.66) | 264.25 (34.94) | *VS* | 38.92 (18.97) | 43.96 (21.28) | 49.89 (25.43) | 61.22 (27.55) |
| *Pre hoc* (*p* value) | NS | NS | NS | 0.008 | *Pre Hoc* (*p* value) | 0.022 | 0.003 | NS | 0.001 |

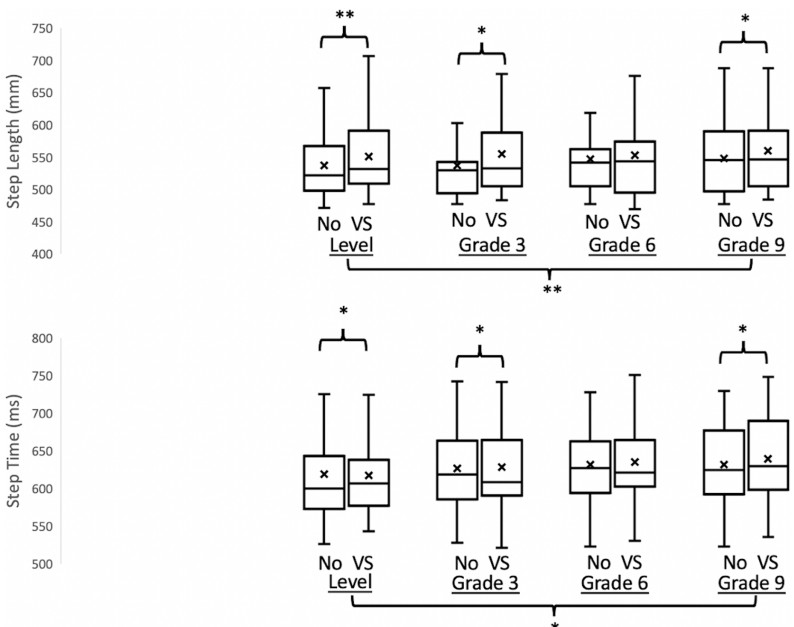

**Figure 3 The results of the step length and the step time with/without vibration-based vestibular stimulation when walking on different inclines.** Empty Box, without vestibular stimulation; Diagonal stripes box, with vestibular stimulation; X, mean values; $^{*}p < 0.05$, $^{**}p < 0.01$.

## DISCUSSION

In this study, we attempted to understand the effect of bilateral VS on gait characteristics during walking on different inclines. This study expected to observe increases in the temporal gait parameters. However, decreases in the spatial gait parameters would also be observed. The results partially agreed with the hypotheses that greater step length and step time were observed at the same time. Additionally, the effect of bilateral VS significantly increased the foot clearance and foot clearance variability.

### Bilateral VS increased the step length and step time but not the step width

Surprisingly, an increase in the step length was observed, specifically, approximately a 3% increment of step length on level, 3%, and 9% incline walking when the bilateral VS was applied in comparison with no VS conditions. Also, this finding diverged from previous studies that patients with vestibular disorders demonstrated smaller step lengths than controls (*Borel et al., 2004*; *Marchetti et al., 2008*). This study provided a possible rationale for explaining this greater step length under bilateral VS as follows. Applying bilateral vestibular stimulation induced an illusion that led participants to sense their body's center of mass is located forward, termed "pressing for forward" (*Ivanenko, Grasso & Lacquaniti, 2000*). *Kavounoudias et al. (1999)* conducted an interesting study that applied single or multiple vibration-based stimulations on neck muscles during standing and found as follows: (1) while applying a single vibration-based stimulation on one side of the neck muscle, the body moved toward the contralateral side, and (2) while the co-vibrations were
applied on adjacent neck muscles orthogonally, the body moved diagonally toward to the contralateral side. Furthermore, *Ivanenko, Grasso & Lacquaniti (2000)* applied the co-vibration on the neck muscles—splenius tendons on both sides symmetrically and found that walking speed increases when walking on a position feedback treadmill (treadmill belt speed is controlled by human walking speed). Both above mentioned studies suggested that applying vibrations on neck muscle induced the illusion. Expressly, *Ivanenko, Grasso & Lacquaniti (2000)* indicated that this illusion might be highly associated with the vibrations that impacted the vestibular system for controlling the body orientation. In other words, when bilateral vestibular systems were perturbed by the VS, participants might feel a forward falling sensation and had no choice but to increase their step length to counterbalance this "press for forward" sensation in the current study. At the same time, because the constant treadmill speed was given to a participant throughout every condition, the step time was forced to be increased with increasing step length to maintain dynamic balance. This result inexplicitly confirmed the illusion induced by VS, like the abovementioned studies, particularly in the anterior-posterior direction. This study speculated that this "pressing for forward" illusion might be associated with nystagmus induced by the bilateral VS (*Manzari & Modugno, 2008*; *Nuti & Mandala, 2005*; *Perez, 2003*); however, the eye movement was not recorded during walking on different inclines in the current study. This speculation needs to be confirmed in future studies.

It is also worth mentioning that the increase in spatial-temporal gait parameters induced by noisy bilateral galvanic vestibular stimulation has been thought of as an improvement in gait in healthy controls and patients with bilateral vestibulopathy (*Iwasaki et al., 2018*). Based on this observation, the supra-threshold Bilateral VS in the current study also had a similar effect on enhancing the gait performance while walking on 0%, 3%, and 9% grades of inclines. On the one hand, applying sub-threshold vestibular stimulation referred to the phenomenon whereby the presence of vibration enhanced the perception of weak sensory stimuli by reducing the threshold of the vestibular system for rehabilitation (*Iwasaki et al., 2018*; *Wuehr, Decker & Schniepp, 2017*; *Wuehr et al., 2022*). On the other hand, applying supra-threshold vestibular stimulation generally has been thought to play a role in perturbing the vestibular system (*Chien, Mukherjee & Stergiou, 2016*; *Lu, Xie & Chien, 2022*). However, in the current study, it was not the case, and applying bilateral VS played a critical key in improving the gait performance while walking on different inclines. "If you want to find the secrets of the universe, think in terms of energy, frequency, and vibration," by Nikola Tesla. Thus, the "benefit" of gait improvement in different locomotor tasks in young adults might depend on various combinations of frequencies and amplitudes. Also, significant increases in spatial-temporal gait parameters were observed while walking on different inclines, inferring that walking inclines might also be a benefit of the gait performance.

Unexpectedly, the VS did not affect the step width at all. This observation was supported by previous studies (*Herssens et al., 2021*). *Herssens et al. (2021)* reported no significant differences in step width during overground walking in patients with bilateral vestibulopathy. This phenomenon could be explained by the "offset" hypothesis. This

hypothesis was observed by *Séverac Cauquil, Gervet & Ouaknine (1998)* that there was no center of pressure (CoP) deviation in the frontal plane when placing galvanic vestibular stimulators on both sides of the mastoid processes during standing. Additionally, this "offset the movement in the medial-lateral and then pressing the forward the movement in the anterior-posterior direction" was supported by *Chien, Mukherjee & Stergiou*'s *(2016)* study during walking. In their study, the effect of bilateral vestibular stimulation didn't increase the net center of pressure sway and respective variability in the medial-lateral (*Chien, Mukherjee & Stergiou, 2016*). Therefore, this study speculated that bilateral mastoid vibration would cancel the perceived "tilt of surface" feeling if both vestibular systems were perturbed simultaneously, resulting in no change in step width but increased deviation in the anterior-posterior direction. To verify this speculation, future studies need to be performed to investigate the movements in the trunk and head.

## VS increased the temporal gait variability but not the spatial gait variability

In the current study, the marginal means revealed the significant VS effect on step time variability but not the step length and width variability. The step time variability increased by 9.6% on level walking, 14% on the 3%, 6.2% on the 6%, and 6.7% on the 9% grade of incline walking. Why did the VS only increase the temporal gait variability but not the spatial gait variability? First of all, increases in gait variability have been thought of as an indicator for continuously identifying self-orientation with the environment in a step-to-step fashion. In the current study, providing the bilateral mastoid vibrations might generate the rhythmical cue for the participant to follow; however, the rhythm of the treadmill was not match up to the rhythm of vibrations. These conflicts in the temporal domain might lead the greater step-to-step fluctuations in step time to adapt this novel locomotor behavior. However, while walking on the constant-speed treadmill, the step length might be relatively easy to be controlled. A similar observation was found that controlling the step-to-step fluctuations in the temporal domain but not the spatial domain might be the key to adapting the novel locomotor behaviors (*Gregory, Sup & Choi, 2021*).

## VS increases the foot clearance and respective variability

Another interesting finding was that the VS increased foot clearance. To our best knowledge, this study was the first to investigate foot clearance while walking on the incline with/without VS. Foot clearance has been suggested as a precise end-point control task during the sway phase (*Winter, 1992*). Importantly, a review concluded that a greater mean of foot clearance and greater foot clearance variability was found in older adults than in young adults and in fallers than in non-fallers. This review further suggested that the increases in foot clearance and respective variability were the safety mechanisms for preventing falls. In the current study, the increases in foot clearance and respective variability could be the results of conflicted-sensory systems. For safety issues, the demands in control of foot clearance become critical by two rationales: (1) for preventing tripping under sensory-conflicted conditions, and (2) for allowing a greater degree of freedom to control foot clearance in the step-to-step fashion. A similar observation was

**Table 2  The effect of bilateral vestibular stimulation on selected gait characteristic during different inclines.** This table demonstrated the effect of bilateral vestibular stimulation on selected gait characteristic from *pre-hoc* while walking on 0%, 3%, 6%, and 9% grades of inclines. Step length (StepLength, mm), step time (StepTime, ms), step time variability (StepTimeV, ms), foot clearance (Footclearance, mm), foot clearance variability (FootclearanceV, mm). No, Walking on the treadmill without bilateral vestibular stimulation; VS, walking on the treadmill with bilateral vestibular stimulation; NS, the *p*-value is larger than 0.05, indicating not significant, NA; the interaction was didn't reach the significance level. The *Post-Hoc* can't be performed. Therefore, NA represented not available.

| StepLength (mm) | 0% of grade | 3% of grade | 6% of grade | 9% of grade | The effect of inclines | The effect of MV | Interaction |
|---|---|---|---|---|---|---|---|
| No | 537.27 (57.27) | 536.82 (54.56) | 546.55 (57.88) | 547.88 (61.44) | $p = 0.001$ | $p < 0.001$ | $p = 0.005$ |
| VS | 551.02 (65.72) | 554.57 (65.94) | 552.42 (65.84) | 559.15 (68.91) | | | |
| *PostHoc* (*p* value) | $p = 0.004$ | $p = 0.014$ | NS | $p = 0.034$ | | | |
| StepTime (ms) | 0% of grade | 3% of grade | 6% of grade | 9% of grade | The effect of inclines | The effect of MV | Interaction |
| No | 619.25 (55.81) | 617.46 (49.98) | 626.74 (59.73) | 628.25 (58.61) | $p = 0.008$ | $p = 0.001$ | $p = 0.028$ |
| VS | 631.39 (57.41) | 635.24 (59.53) | 632.09 (59.36) | 639.64 (58.15) | | | |
| *PostHoc* (*p* value) | $p = 0.027$ | $p = 0.028$ | NS | $p = 0.028$ | | | |
| StepTimeV (ms) | 0% of grade | 3% of grade | 6% of grade | 9% of grade | The effect of inclines | The effect of MV | Interaction |
| No | 15.89 (4.40) | 15.29 (4.88) | 17.02 (5.44) | 17.23 (6.31) | NS | $p = 0.024$ | NS |
| VS | 17.41 (4.35) | 17.49 (4.75) | 18.07 (4.89) | 18.37 (6.27) | | | |
| *PostHoc* (*p* value) | NA | NA | NA | NA | | | |
| FootClearance (mm) | 0% of grade | 3% of grade | 6% of grade | 9% of grade | The effect of inclines | The effect of MV | Interaction |
| No | 231.98 (30.79) | 243.32 (33.02) | 253.18 (32.94) | 260.40 (34.52) | $p < 0.001$ | $p < 0.001$ | NS |
| VS | 235.01 (32.22) | 246.48 (30.59) | 256.58 (32.66) | 264.25 (34.94) | | | |
| *PostHoc* (*p* value) | NA | NA | NA | NA | | | |
| FootClearanceV (mm) | 0% of grade | 3% of grade | 6% of grade | 9% of grade | The effect of inclines | The effect of MV | Interaction |
| No | 35.96 (20.12) | 39.32 (20.05) | 44.75 (23.52) | 51.98 (28.11) | $p < 0.001$ | $p < 0.001$ | $p = 0.003$ |
| VS | 38.92 (18.97) | 43.96 (21.28) | 49.89 (25.43) | 61.22 (27.55) | | | |
| *PostHoc* (*p* value) | NS | $p = 0.003$ | NS | $p = 0.001$ | | | |

found in *Ivanenko, Grasso & Lacquaniti*'s *(2000)* study; participants elevated the average thigh-shank-foot loop while the vibrations were activated on bilateral neck muscle. Again, *Ivanenko, Grasso & Lacquaniti (2000)* suggested this elevation of kinematics was evoked by illusion. Therefore, the third rationale for raising the foot clearance and respective variability was to counterbalance the VS-induced illusion.

## Why did the VS not affect the step length and step time while walking on the 6% grade incline?

Unexpectedly, the VS did not affect the step length and step time while walking on the 6% grade incline. When inspecting the step length and step time (Table 2), the step length had a sharp increase while walking without VS in both step length and step time from a 3% grade incline to a 6% grade incline, implying the possible alternations in gait pattern when walking from 3% to 6% grade of inclines. The rationale might be that, on the one hand, the locomotor tasks became a bit challenging as the grade of incline increased. Thus, participants had to sharply increase their step length and step time to adapt to this new locomotor task even though no bilateral VS was applied. On the other hand, walking with bilateral VS itself was already challenging; therefore, there were no apparent changes from

3% to 6% grade of inclines. Thus, the step length and time under bilateral VS were similar to the ones under no bilateral VS while walking on a 6% grade of incline. This phenomenon was also observed by Earhart and Bastian that healthy young participants used different control mechanisms when they stepped on steeper grades of a wedge (*Earhart & Bastian, 2000*). Similarly, another study, which investigated the first step on an inclined surface, suggested the above observation that the control mechanisms were changed to prepare the limb for an elevated heel contact with increased propulsive force when the grade of inclines reached a certain level (*Prentice et al., 2004*). In short, this study proposed an explanation for this phenomenon: between 3% to 6% grade incline might be a transient phase from one control mechanism to another for gait.

## CONCLUSIONS

In conclusion, walking with VS increased step length, step time, and foot clearance. Surprisingly, while walking on a certain grade of incline, the effect of VS on spatial-temporal gait parameters might be offset due to the transient phase from one control mechanism to another control mechanism. This study suggested that step length, step time, foot clearance, step time variability, and foot clearance variability might be the primary parameters for future clinic diagnoses for patients with BVD while walking on different inclines. Also, the increases in spatial-temporal gait parameters might be a positive indicator for gait improvement while the bilateral VS was applied during different inclines walking.

### Limitations

The apparent limits of this study were the lack of healthy older and pathological groups. To our best knowledge, this exploratory study was the first study to investigate the effect of perturbed vestibular systems on gait characteristics while walking on different inclines. Many gait characteristics were investigated in this study to prioritize the importance of each gait characteristic. Therefore, only healthy young adults were recruited for the study for clear descriptions and to prevent potential statistical problems. Further research could compare the walking performance of healthy subjects with VS and patients with vestibulopathy in different settings. Another apparent limit was a small sample size.
To solve this problem, the *pre-hoc* tests were applied to prioritize the importance of each gait characteristic. Five out of twelve gait characteristics were selected for future statistical analysis for compensating statistical procedure shortcomings. Although the sample size was relatively small in the current study, the effect size demonstrated a large effect. These outcomes of large effect size also were similar to previous studies, which recruited twenty healthy young adults and investigated more than ten gait parameters (*Chien, Mukherjee & Stergiou, 2016*; *Lu, Xie & Chien, 2022*). Additionally, the statistical power was calculated by GPower, and the 80% of statistical power, which represented a reliable balance between the alpha and beta risk, could be reached by recruiting 18 healthy young adults for the repeated measure. Another limitation was that the eye tracker device was not used in the current study. Thus, the abnormal eye movement induced by bilateral VS could not be confirmed. This limitation needs to be performed in future research.

## ACKNOWLEDGEMENTS

We would like to thank all participants for their contribution to the study. All data collection was performed at the Clinical Movement Analysis Lab, Department of Health & Rehabilitation Science at the University of Nebraska Medical Center. We sincerely thank the generosity of the Department of Health & Rehabilitation Science for using the equipment.

### Funding
The authors received no funding for this work.

### Competing Interests
Jung H Chien is an independent researcher and is not employed by any companies. Yuxiao Sun and Dongqi Zhu are physical therapy students in University of Nebraska Medical Center. Huiyan Song is an employee of West China Hospital in China. All authors declare that they have no competing interests.

### Author Contributions
- Yuxiao Sun conceived and designed the experiments, analyzed the data, prepared figures and/or tables, authored or reviewed drafts of the article, and approved the final draft.
- Dongqi Zhu conceived and designed the experiments, authored or reviewed drafts of the article, and approved the final draft.
- Huiyan Song conceived and designed the experiments, authored or reviewed drafts of the article, and approved the final draft.
- Jung H. Chien performed the experiments, prepared figures and/or tables, authored or reviewed drafts of the article, and approved the final draft.

### Human Ethics
The following information was supplied relating to ethical approvals (*i.e.*, approving body and any reference numbers):

The University of Nebraska Medical Center Institutional Review Board approved this study (IRB# 379-17-EP).

### Data Availability
The raw data is available in the Supplemental Files.

### Supplemental Information
Supplemental information for this article can be found online at http://dx.doi.org/10.7717/peerj.15111#supplemental-information.

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
