# Peer review of "Vibrations on mastoid process alter the gait characteristics during walking on different inclines"

_PeerJ, doi:10.7717/peerj.15111_

## Round 0.1 · original submission · Major Revisions

Line 144 – What do you meant by “quick standing”?

Need to provide more details on how the heel strike, stride length, and step width were determined from the kinematic data.

I agree with the reviewer’s comment about the need to provide details on how the Bonferroni adjustments were done and the actual adjusted p values for the post hoc comparisons.

If an interaction is significant, the significant main effects should not be analyzed. Instead, the authors should focused on analyzing the post hoc comparisons.

Reviewer 1 ·

Basic reporting

The authors performed a study investigating the influence of vibration-based stimulations (VS) while walking on different inclines on a treadmill in 19 young and healthy adults aged bout 24 years. The authors report an increase of step length, step time, and foot clearance variability when walking on 0%, 3%, and 9% incline and an increased the step length and time at 9%.
The topic of the manuscript is very interesting, but there is some space for improvement.

In the basic reporting section, the main comments is, that the mechanisms of the effect of vestibular vibrations is not discussed but is probably important for a deeper understand in interpretation of the results. The intention of the vibration based stimuation is here unclear (simulation a damage? purpose of rehabilitation?)
Additionally the introduction creates an expection of an investagtion of patients with bilateral vesibular dysfunction of with falls.

Experimental design

The experimental design has considerable deficiencies:

Methods:
Including only healthy young adults is major limitation of the study. It seems to be better to include two further cohorts (healthy older adults and older adulty with bilateral vestibular dysfunction). With these two cohorts more questions of the introduction could have been answered and the study would be more clinically relevant.

The number of participants is very low. Comparing 12 gait parameters in each 4 conditions = 48 parameters, in a cohort of 19 participants might cuase statisitical problems.

The statistical procedure is not repeatable as important details were not reported, e.g. the “pre hoc” procedure is not described.

The described statistic procedure has some shortcomings:
For post hoc testing between different conditions (VS vs. no VS) in the same cohort paired tests should be used.
l193 the Bonferroni correction should be described in more detail. Due to the high number of parameters it should also be applied to the “pre hoc” testing.

Which threshold for significance was chosen after correction?

The criteria, why NA for p-values in the tables is not clear.

Validity of the findings

Raw data are available, but despite of the statistical shorcomings reported above there are further deficiencies in the interpretation of the results:

An important part in the discussion is about a decrease of double support time and stance time under stimulation at 9%. This decrease is not based on the data presented in table 1.

There are significant values for the effect of the stimulation over all conditions but there are not p-values are NA in for the direct comparision (VS vs No VS). (foot clearand and variabtiliy of step width)

The missing significant results at 6% are strange and not explained in the discussion

The paragraphs 318ff and 350ff are rather speculative, especially the recommendation for diagnostic and rehabilitation, because both aspects are not investigated in the current study.

Conclusion is also very speculative and not a real conclusion of the study.

Additional comments

Minor comments:

Abstract
A reason why a vibration-based stimulation at the mastoid makes sense should be given in the abstract
The experimental setup is difficult to understand in the abstract
The method of measuring gait needs to be reported
Conclusion 2 is difficult to follow with the information given in the abstract

Introduction
Why is this limited to bilateral dysfunction? A bilateral but asymmetric damage is bilateral but might behave like a unilateral hypofunction.

46-48 Is a bilateral vestibular damage really rare? Especially in older adults it seems to be quite frequent and probably underdiagnosed.

Long sentences e.g. 73-77 are difficult to understand

L 78 the number of study participants and their age should be given, as a possible explanation of the missing significance is the low number.

Methods:
Was there a check, that participants did not had an unknown vestibular damage?
Age range should be reported

The exact position of the markers should be shown in a figure.

175 what was the rational to use SD and not CoV for variation measures? (both war possible, but a rational for the choice would be great)

Why was the frequency of 100hz used for stimulation and not 30 or 60hz, as used for the SVINT?

·

Basic reporting

The authors provide a comprehensive explanation of the role of the vestibular system, specifically addressing falls as a consequence of bilateral vestibular hypofunction. Some statements should be adjusted from a clinical neurological perspective: In line 39, the authors claim that any vestibular system hypofunction results in nystagmus, gait unsteadiness, and limitations in spatial orientation. In general, one should be very careful with generalized statements in medical questions. It is wrong that every vestibular hypofunction leads to nystagmus. This is typical for an acute unilateral vestibulopathy, but in bilateral vestibulopathies usually no nystagmus is observed (Strupp M, Kim JS, Murofushi T, Straumann D, Jen JC, Rosengren SM, Della Santina CC, Kingma H. Bilateral vestibulopathy: Diagnostic criteria Consensus document of the Classification Committee of the Bárány Society. J Vestib Res. 2017;27(4):177-189. doi: 10.3233/VES-170619. PMID: 29081426; PMCID: PMC9249284). Similarly, orientation in space need not be impaired in every case. I also recommend to be very careful with the statement that every hypofunction leads to gait instability. In any case, the literature source cited does not support this statement.

Line 51 reports that an objective measurement for bilateral vestibular hypofunctions should be developed. In this regard, there are clinical examinations, as well as video-oculography and caloric testing, which are applied as standards and are easy to perform. If the authors see the method as a future diagnostic method, they should also address the added value compared to previous methods.

Experimental design

The authors use vestibular stimulation and different gradients to investigate whether vestibular stimulation as a function of gradient leads to a change in gait pattern, studied on different parameters.

The authors did not clearly emphasize whether vestibular stimulation should cause a disturbance of the vestibular system or cause an improvement, as is the case with galvanic stimulation (Keywan A, Wuehr M, Pradhan C, Jahn K. Noisy Galvanic Stimulation Improves Roll-Tilt Vestibular Perception in Healthy Subjects. Front Neurol. 2018 Mar 1;9:83. doi: 10.3389/fneur.2018.00083. PMID: 29545766; PMCID: PMC5837962.). Even if the effect of vestibular stimulation is unclear, this should be stated in the clearest terms. Although the authors describe changes in gait pattern that indicate a deterioration of vestibular functions, this should also be clearly formulated.

In my opinion, the research question is reasonable, even if further studies will be necessary. Altering the vestibular system by vestibular stimulation can be used to assess vestibular disruption at healthy participants.

Validity of the findings

In essence, the study supports previously published results cited in this paper. The results for the dependence on slope showed no significant result for many parameters. However, stride length is a parameter that may be altered by slope and by stimulation, was even increased in this study. It remains unclear what the significance of the increase in step length is and whether this will remain reproducible in further studies, since this effect is unlike other studies cited by the authors. The authors give a possible explanation, that chronic vestibular dysfunction may have a different pattern of altered gaitparameters. It should be discussed that vestibular stimulation rather indices a pattern of chronic vestibular dysfunction and a possible reason should be given.

Additional comments

The authors provide an overview of the role of vestibular stimulation on different types of inclines and observed an increase in the effect of vestibular stimulation on at least some parameters. Further studies will be necessary, but this work provides new aspects for a better understanding of the orchestration of the vestibular system and locomotion.

---

## Round 0.2 · Minor Revisions

This revision has addressed the comments made by the reviewer and is almost ready for publication.

Please address the attached edits from the Section Editor.

·

Basic reporting

The authors provide a good overview of the function bilateral vestibular dysfunction (BVD) and the role of vibration induced stimuli (VS). Alternatively used galvanic stimulation is also explained and the contrast to noisy galvanic stimulation as a possible improvement of the vestibular apparatus. Previously established clinical methods are also explained and discussed. In summary, a differentiated and from a clinical point of view also reasonable evaluation takes place.

Experimental design

In this version, there is a clearly understandable treatise on the function of the vestibular apparatus and on the effects at different slopes. In line 699, the sentence "the interaction was didn’t reach the significance level" should be without the word "was".

Validity of the findings

There is a clinically plausible discussion of the role of the vestibular apparatus and possible explanations of the findings.

Additional comments

As a clinician, I am particularly interested in publications that provide new insights into disease patterns. Even though bilateral vestibular hypofunction is described as rare, I perceive it frequently in patients, at least subjectively, although not always to its maximum extent. The authors provide an interesting discussion of further research approaches that I look forward to.

---

## Round 0.3 · accepted · Accept

Thank you for addressing the Section Editor and reviewer comments.